# Therapeutic Applications of Botulinum Neurotoxin for Autonomic Symptoms in Parkinson’s Disease: An Updated Review

**DOI:** 10.3390/toxins13030226

**Published:** 2021-03-19

**Authors:** Steven D. Mitchell, Christos Sidiropoulos

**Affiliations:** Department of Neurology, Michigan State University, East Lansing, MI 48824-7015, USA; sidirop3@msu.edu

**Keywords:** autonomic, botulinum neurotoxin, botulinum toxin, non-motor, Parkinson’s disease

## Abstract

Parkinson’s disease is the most common age-related motoric neurodegenerative disease. In addition to the cardinal motor symptoms of tremor, rigidity, bradykinesia, and postural instability, there are numerous non-motor symptoms as well. Among the non-motor symptoms, autonomic nervous system dysfunction is common. Autonomic symptoms associated with Parkinson’s disease include sialorrhea, hyperhidrosis, gastrointestinal dysfunction, and urinary dysfunction. Botulinum neurotoxin has been shown to potentially improve these autonomic symptoms. In this review, the varied uses of botulinum neurotoxin for autonomic dysfunction in Parkinson’s disease are discussed. This review also includes discussion of some additional indications for the use of botulinum neurotoxin in Parkinson’s disease, including pain.

## 1. Introduction

First described in 1817 by James Parkinson in his work “An Essay on the Shaking Palsy” [1], Parkinson’s disease (PD) is the most common age-related motoric neurodegenerative disease [2,3,4]. Estimates indicate that in 2010 the prevalence of PD in those over age 45 was 680,000 in the United States, with that number estimated to rise to 1,238,000 in 2030 [5]. The four cardinal features of PD are resting tremor, rigidity (i.e., increased resistance throughout the range of passive limb movement), bradykinesia (i.e., slowness of movement), and postural instability [6,7,8]. The non-motor aspects of PD include a variety of domains such as sleep disorders, pain, cognitive dysfunction, and autonomic dysfunction (i.e., dysautonomia) [2,7,9].

Idiopathic PD has traditionally been considered the most common form of parkinsonism [7,10,11]. More recently, however, the concept of idiopathic PD as a single entity has been challenged [7,8,12]. As we learn more about clinical subtypes, pathogenic genes, and possible causative environmental agents of PD, it seems logical that PD is more diagnostically complex than initially thought [7,12,13].

Pathologically, PD is associated with Lewy bodies and Lewy neurites consisting of misfolded and aggregated alpha-synuclein protein [7,14,15]. Physiologically, PD results in loss of dopamine neurons in the substantia nigra pars compacta and subsequent basal ganglia dysfunction [4,15], as well as reduction in mitochondrial activity [4,7,16]. In addition, it has been found that disruption of non-dopaminergic pathways (e.g., noradrenergic, glutamatergic, serotonergic, and adenosine pathways) also occurs in PD and may account for the various non-motor symptoms [7,17].

Carbidopa-levodopa is considered the gold standard treatment for PD and mainly acts by replenishing dopamine in the nigrostriatal pathway [2,7,18]. Carbidopa-levodopa improves motor symptoms to a variable extent but does not substantially affect non-motor aspects of the disease [7,19,20]. With prolonged use, patients become less responsive to dopaminergic agents and experience a narrowing therapeutic window with side effects such as motor fluctuations, dyskinesias, autonomic nervous system dysfunction, and various neuropsychiatric symptoms [7,18,21]. Alternative therapies for PD include dopamine agonists, anticholinergics, monoamine oxidase inhibitors (MAOIs), catechol-O-methyl transferase inhibitors (COMTIs), deep brain stimulation, focused ultrasound lesioning, and botulinum neurotoxins [2,7,8,22].

Botulinum neurotoxin (BoNT), regarded as the most potent toxin known to mankind, is produced by the anaerobic spore-forming Gram-positive bacillus *Clostridium botulinum* [23,24,25]. BoNT is a zinc protease that cleaves neuronal vesicle-associated proteins responsible for acetylcholine release into the neuromuscular junction [23]. See Figure 1 for a depiction of the mechanism of action. BoNT was first approved for medical use in Canada in the late 1980s for the treatment of strabismus (i.e., eye misalignment) [23]. FDA approval for clinical use of BoNT in the United States for strabismus and blepharospasm was subsequently granted in 1989 [23]. Since that time, the indications and use of BoNT have expanded.

**Figure 1 toxins-13-00226-f001:**
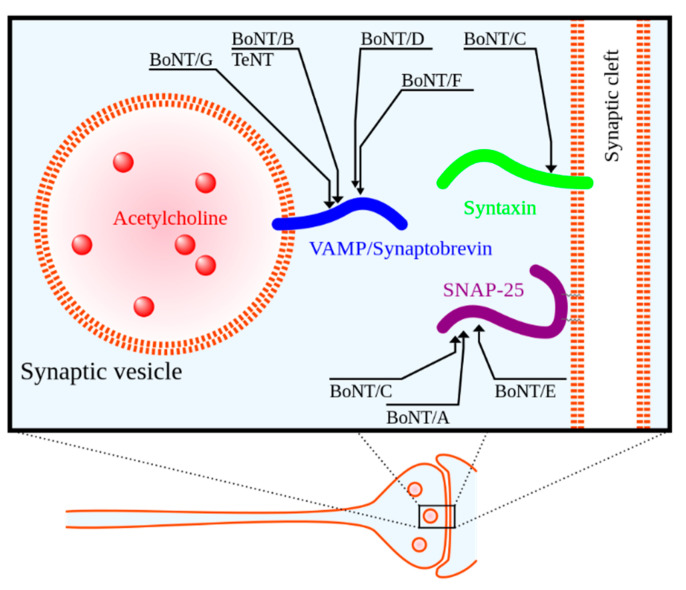
Molecular targets of clostridial neurotoxins in presynaptic cell. BoNT/A–G = botulinum toxin serotypes A–G, TeNT = tetanus toxin. Reproduced from Wikipedia Commons. Adapted by Y tambe from [26]. 2005, *Emerg. Infect. Dis*.

There are seven biological serotypes of BoNT, with two of these serotypes used in clinical practice [24,25]. BoNT serotypes A and B, of which there are four formulations currently on the market, are used for a variety of medical indications including glabellar lines (i.e., facial wrinkles), dystonia (a disorder of involuntary muscle contractions causing repetitive or twisting movements), spasticity, and migraines [24,25,27,28]. See Table 1 for a summary of the four BoNT formulations currently available.

This systematic review aims to serve as an up-to-date summary of the varied uses of BoNT for autonomic dysfunction in PD, including sialorrhea, hyperhidrosis, gastrointestinal dysfunction, and urinary dysfunction. Our discussion will also review some additional indications for the use of BoNT in PD, including pain.

## 2. Results

Duplicate articles were removed, and remaining articles were screened for relevance and subsequently categorized by specific topic. See Figure 2 for a summary of study selection.

**Figure 2 toxins-13-00226-f002:**
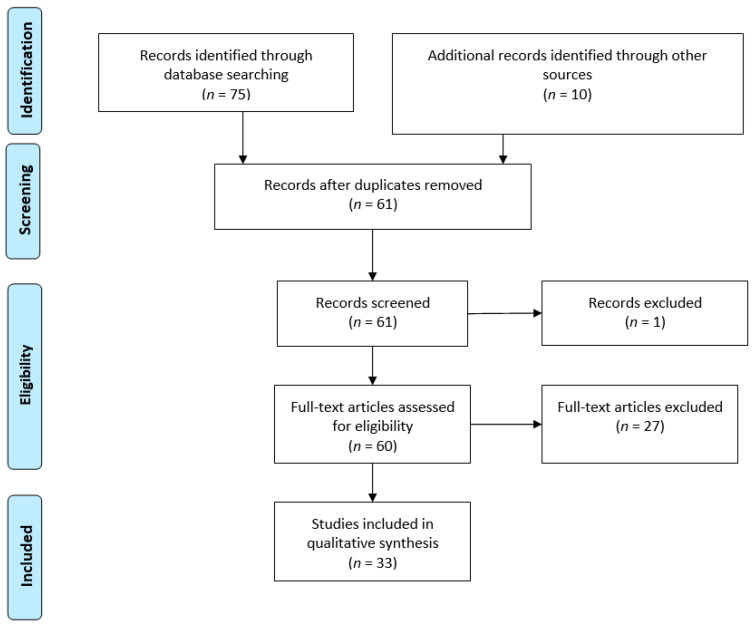
PRISMA flow diagram of studies included in the review. Reproduced from [29]. 2009, *PLoS Med*.

The majority of studies (*n* = 26) were reviews, while the remaining (*n* = 7) consisted of retrospective cohort studies (*n* = 2), case report (*n* = 1), expert opinion (*n* = 1), cross-sectional study (*n* = 1), and randomized controlled trials (*n* = 2). See Table 2 for a summary of study findings.

**Table 2 toxins-13-00226-t002:** Summary of study findings included in this review.

Topic	Lead Author (Year) [Reference]	Type of Study	Toxin Type + Total Patients (If Applicable)	Result/Conclusion
Sialorrhea	Tiigimäe-Saar (2017) [30]	Retrospective cohort	abobotulinumtoxinA 38 (12 injected with BoNT via US guidance)	Significant decrease in salivary flow rate was observed in those treated with BoNT type A. No significant change was found in salivary composition at 1 month follow-up.
	Hayes (2019) [31]	Review	A + B	Off-label BoNT injection into parotid or submandibular glands (with US guidance) can be an effective treatment for sialorrhea.
	Papesh (2019) [32]	Case report	onabotulinumtoxinA + rimabotulinumtoxinB 1	Case report of parotitis and sialolithiasis following BoNT injection.
	Seppi (2019) [33]	Review	A + B	BoNT type B is efficacious and clinically useful for drooling based on one high-quality positive study. Generally, BoNT type A and type B are considered to pose an acceptable risk with specialized monitoring techniques.
	Kulshreshtha (2020) [34]	Review	A	BoNT is the best treatment for management of sialorrhea, but repeated injections are needed.
	Quarracino (2020) [35]	Expert opinion	A + B	Sialorrhea can be managed with BoNT injections and oral glycopyrrolate.
Gastrointestinal Dysfunction	Triadafilopoulos (2017) [36]	Retrospective open cohort	not specified 14	Endoscopic BoNT injection to the esophagus, pylorus or anal canal is safe, well-tolerated, and leads to symptomatic improvement of dysphagia, gastroparesis, and chronic anismus that lasts up to several months.
	Ramprasad (2018) [37]	Review	A	BoNT type A can prove beneficial for patients who fail levodopa treatment and display significant upper esophageal sphincter dysfunction.
	Sharma (2018) [38]	Review	not specified	Chronic constipation is a common, nonmotor, and prodromal symptom in PD. Its underlying neuropathology may provide pathophysiological insight into PD. BoNT injection into the puborectalis may help dyssynergic defecation.
	Barbagelata (2019) [39]	Review	A	BoNT type A for dysphagia may be an effective and safe alternative to invasive procedures or may be useful to identify patients who might benefit from surgical myotomy.
	Mendoza-Velásquez (2019) [40]	Review	not specified	BoNT in the distal esophagus could improve dysphagia.
Urinary Dysfunction	Brucker (2017) [41]	Review	onabotulinumtoxinA + abobotulinumtoxinA	BoNT can be used for intractable urinary incontinence in PD with a risk of impairing bladder emptying.
	Madan (2017) [42]	Review	A	Intravesical BoNT has been shown to be effective in the management of urinary symptoms in PD patients, and there is evidence that it may improve detrusor sphincter dyssynergia.
	McDonald (2017) [43]	Review	A	Intravesical BoNT may be helpful for lower urinary tract symptoms in PD, but trials have been small and frequently lacked a suitable control group, making them vulnerable to placebo effect. Any patient undergoing intra-detrusor BoNT must be consented for possible urinary retention and need for an intermittent or indwelling catheter.
	Sakakibara (2018) [44]	Review	A	Intramural, multiple BoNT injection in the bladder seems to be a promising method to treat intractable detrusor overactivity in patients with PD.
	Mehnert (2019) [45]	Review	A	BoNT type A intra-detrusor injections are a safe treatment with few adverse events. BoNT type A intra-detrusor injections are recommended as second line treatment for neurogenic detrusor overactivity refractory to antimuscarinic treatment.
	Miller-Patterson (2020) [46]	Cross-sectional	not specified 308	Men with PD are more likely than women to receive a medication, such as BoNT, for urinary symptom treatment, despite no difference in overall urinary symptom prevalence.
Pain	Bruno (2017) [47]	Randomized controlled crossover trial	A 12	BoNT for dystonic and musculoskeletal pain showed a mild, non-significant reduction in pain after 4 weeks compared to placebo.
	Rana (2017) [48]	Review	not specified	BoNT has been shown to alleviate painful dystonias associated with PD, including blepharospasm, axial dystonia, camptocormia, and limb dystonia.
	Rieu (2018) [49]	Randomized controlled trial	incobotulinumtoxinA 45	BoNT injections are effective for improving clinical state of parkinsonian patients with plantar flexion of toe dystonia.
	Buhidma (2020) [50]	Review	not specified	BoNT can be used for dystonic pain in PD. Treatment of pain in PD remains less than optimum. Rodent models may assist with better understanding the mechanism of PD-related pain.
	Buhmann (2020) [51]	Review	A	Off-label injection of BoNT into pain trigger points might be helpful. BoNT can also help with painful myotonus or spasms in the esophagus, pylorus, anal sphincter, or in painful detrusor hyperactivity of the bladder.
	Karnik (2020) [52]	Review	A	BoNT type A had a non-significant signal toward improving dystonic limb pain in PD.
	Tai (2020) [53]	Review	A	Pain is a heterogeneous symptom in PD. A randomized double-blind crossover study tested BoNT type A for limb pain in advanced PD. This treatment did not significantly reduce the pain score in the pain group; however, a subgroup analysis showed that it significantly improved dystonic pain.
Multiple Topics	Jost (2017) [54]	Review	not specified	BoNT can be considered for use in focal hyperhidrosis. BoNT is also approved for the treatment of neurogenic and idiopathic detrusor hyperactivity and should be considered for therapy-resistant cases.
	Shukla (2017) [27]	Review	A + B	BoNT can effectively ameliorate the symptoms of cervical dystonia, blepharospasm, sialorrhea, and hyperactive bladder and is increasingly being used for additional PD-related indications including limb dystonia, oromandibular dystonia, tremors, constipation, dysphagia, gastroparesis, and sweating dysfunction.
	Sławek (2017) [55]	Review	A + B	BoNT offers effective treatment for drooling and bladder dysfunctions and alternative treatment for constipation and pain related to parkinsonism.
	Safarpour (2018) [56]	Review	A + B	Literature supports a level A efficacy (established) for BoNT therapy in cervical dystonia and a level B efficacy (probably effective) for blepharospasm, hemifacial spasm, laryngeal dystonia (spasmodic dysphonia), task-specific dystonias, essential tremor, and PD rest tremor.
	Tater (2018) [57]	Review	A + B	BoNT is effective for apraxia of eyelid opening with blepharospasm in PD. BoNT is also effective for sialorrhea, axillary hyperhidrosis, detrusor overactivity, and pain in PD.
	Jocson (2019) [58]	Review	A + B	Uses of BoNT in idiopathic PD include sialorrhea, limb, dystonia, tremor, dyskinesias, freezing of gait, camptocormia, pisa syndrome, urinary dysfunction, constipation, dysphagia, eyelid opening apraxia, and blepharospasm.
	Chen (2020) [59]	Review	A + B	BoNT is effective for treatment of sialorrhea in PD. In addition, BoNT type A may improve gastroparesis and defecatory dysfunction in PD. BoNT type A has been demonstrated to be effective for the treatment of urinary symptoms in PD.
	Jankovic (2020) [7]	Review	not specified	BoNT is effective in controlling high-amplitude rest and postural hand tremor, which may be resistant to levodopa. BoNT may be also beneficial in the treatment of a variety of other non-levodopa responsive parkinsonian symptoms such as blepharospasm, apraxia of eyelid opening, anterocollis, camptocormia, bruxism, sialorrhea, seborrhea, hyperhidrosis, overactive bladder, and constipation.
	Savitt (2020) [60]	Review	not specified	Blepharospasm responds to BoNT injection of the eyelids.

Abbreviations: A = toxin type A, B = toxin type B, BoNT = botulinum neurotoxin, PD = Parkinson’s disease, US = ultrasound.

## 3. Discussion

### 3.1. Botulinum Neurotoxin for Sialorrhea

Sialorrhea, otherwise known as excessive salivation or drooling, is a common non-motor symptom seen in approximately 50–70% of PD patients [34,55]. Sialorrhea is likely multifactorial and is thought to occur from either increased saliva secretion or inadequate clearance [27]. Dysphagia, or difficulty swallowing, can contribute to inadequate clearance and is likely the primary mechanism of drooling in PD [55]. In addition, patients with more advanced stages of PD often exhibit a flexed neck posture, which in conjunction with an open jaw can contribute to drooling [27,55]. Excessive drooling can unfortunately lead to social embarrassment and worsening depression [35], as well as poor oral and perioral hygiene and respiratory tract infections [55].

Ultrasound (US)-guided injection of BoNT into parotid and submandibular glands can be recommended as first line treatment for sialorrhea, especially when anticholinergic oral medications are not indicated due to the risk of confusion, cognitive decline, or psychosis [31,55]. Oral glycopyrrolate, a muscarinic anticholinergic, can be used for sialorrhea [35], however, long-term studies on it are lacking [34]. Sublingual atropine 1% has also been suggested as an alternative therapy for sialorrhea in PD [61], however, no randomized controlled trials have been carried out.

Several recent studies have confirmed the efficacy of BoNT for sialorrhea [33,62,63]. A small study in 2018 showed efficacy of incobotulinumtoxinA for reducing sialorrhea in children with post-anoxic cerebral palsy [62]. An earlier retrospective cohort study of 45 neurologically impaired children also found that US-guided BoNT type A injections into the salivary glands was safe and efficacious for drooling, with an approximately 5 month mean duration of effect [63] and onset of action approximately 1 week following BoNT injection [55]. In a systematic review by Seppi and colleagues, BoNT type A and type B were both deemed efficacious and with acceptable risk on the basis of well-designed randomized clinical trials for the treatment of drooling in PD patients when administered by well-trained physicians with specialized monitoring techniques, such as US guidance [33].

As with any invasive procedure, infection, hematoma, and pain are known risks of BoNT injection [32]. A single case report of parotitis and sialolithiasis (i.e., salivary gland stones) following injection with rimabotulinumtoxinB was found in our literature review [32]. In addition, BoNT injection into the parotid and submandibular glands may lead to transient dysphagia [55] or xerostomia (i.e., dry mouth) [58]. Nevertheless, a study conducted in 2017 by Tiigimäe-Saar and colleagues indicates that use of BoNT (particularly BoNT type A) can effectively slow salivary flow rate without changing the salivary composition. This indicates that BoNT can effectively treat sialorrhea without impacting the oral health of the patient [30].

### 3.2. Botulinum Neurotoxin for Hyperhidrosis

Hyperhidrosis, defined as excessive sweating, has been reported in 65% of patients with PD [57]. Those PD patients with chronic hyperhidrosis tend to have a higher burden of autonomic symptoms [59,64] as well as higher dyskinesia scores, higher depression and anxiety, and worse quality of life [64]. Hyperhidrosis is usually related to off states or to severe dyskinesia with high energy output [55]. Carbidopa-levodopa and dopamine agonists, two of the most common classes of medications used to treat PD, unfortunately have sweating as a potential side effect as well [7].

Recent reviews highlight the fact that no randomized controlled trials on the use of BoNT for hyperhidrosis in PD have been performed [33,55,57]. Nevertheless, the use of BoNT type A for the treatment of axillary hyperhidrosis has a level A (i.e., established as effective) recommendation [25,27,55,57]. Following local anesthesia, intradermal BoNT injections of the axilla may be performed. By blocking acetylcholine release from sympathetic nerve fibers, BoNT effectively denervates eccrine sweat glands [55]. Treatment is effective for up to 9 months [55]. It is important to note that hyperhidrosis in PD is usually generalized in nature [55], and given the focal nature of BoNT injections, this treatment will only be effective in regions treated.

### 3.3. Botulinum Neurotoxin for Gastrointestinal Dysfunction

Gastrointestinal (GI) dysfunction in PD includes dysphagia (i.e., difficulty swallowing), gastroparesis (i.e., delayed gastric emptying), and constipation [37,40,55,58]. Some have proposed that abnormal accumulations of alpha-synuclein in the periphery may account for the GI dysfunctions observed in PD patients [37,40,65], and the much-debated Braak hypothesis proposes that sporadic PD originates from Lewy pathology in the GI tract [66]. BoNT can be useful for dysphagia, gastroparesis, and constipation.

Dysphagia may lead to aspiration and subsequent pneumonia [39], which unfortunately is a leading cause of death in PD [67]. Dysphagia may be classified as either oropharyngeal or esophageal in etiology [37]. Oropharyngeal dysphagia is estimated to occur in up to 80% of PD patients, while esophageal dysphagia is probably less common [37]. BoNT may be useful for both types [36,37,58]. BoNT injection of the cricopharyngeal muscle has shown benefit for oropharyngeal dysphagia, as evidenced by videofluoroscopy or high-resolution pharyngeal manometry [37,58], while BoNT injection of the lower esophagus and esophago-gastric junction has shown benefit for esophageal dysphagia [36,37]. Benefits from cricopharyngeal muscle injection can last 16–20 weeks [58], and favorable response to percutaneous injection of BoNT type A into the cricopharyngeal muscle may be a useful tool in identifying patients who could benefit from surgical myotomy [39].

Gastroparesis in the form of nausea and vomiting occurs in approximately 25% of PD patients, while abdominal bloating is reported in up to 45% of PD patients [37]. Gastroparesis may not only affect nutrition but also secondarily cause delayed absorption of oral levodopa leading to worsening of motor fluctuations [37,55]. Endoscopic injection of 100–200 units of BoNT into the pylorus of nine PD patients by Triadafilopoulos and colleagues resulted in subjective improvement in early satiety, bloating, epigastric pain, and nausea [36]. As noted by Sławek and colleagues, earlier studies using onabotulinumtoxinA and incobotulinumtoxinA injection of the pylorus also showed possible benefit in a small number of PD patients, but no double-blinded randomized controlled studies have confirmed these findings [55].

Constipation affects approximately 60% of PD patients [37,58]. Constipation in PD likely occurs from either slow transit, outlet obstruction from focal dystonia of pelvic floor muscles, or a combination of both [37,38,58]. Importantly, constipation is considered a prodromal symptom in PD and may predate onset of motor symptoms by up to 16 years [59,68]. As noted in a review by Jocson and colleagues, two small open label studies in the early 2000s suggested that onabotulinumtoxinA injection of the puborectalis muscle is effective for constipation from outlet-type obstruction [58]. The study by Triadafilopoulos and colleagues included two patients who received BoNT injection of the anal sphincter or puborectalis muscle, one with improvement in constipation from paradoxical anal contraction and the other with improvement in constipation from slow transit [36]. Fecal incontinence is a potential side effect of BoNT injection into these muscles [55].

### 3.4. Botulinum Neurotoxin for Urinary Dysfunction

Prevalence of urinary dysfunction in PD patients is estimated to be as low as 35% [57] or as high as 64–67% [58,59]. Lower urinary tract symptoms include both dysfunction of urinary storage and urinary voiding [43]. Storage symptoms include urgency, frequency, and nocturia (i.e., frequent urination at night), while voiding symptoms include slow or interrupted stream, terminal dribble, hesitancy, and straining [34,43]. Neurogenic overactive bladder manifesting as nocturia is the most reported urinary symptom in PD patients [55,58]. Complications from urinary tract dysfunction include upper urinary tract damage and recurrent urinary tract infections [45]. Detrusor muscle overactivity and detrusor-sphincter dyssynergy are frequently implicated [27,44,45].

OnabotulinumtoxinA is an effective and approved treatment for neurogenic detrusor overactivity when antimuscarinic medications are not effective or cause side effects [41,42,45,54]. Intradetrusor injection with up to 360 units of onabotulinumtoxinA may be administered as frequently as every 3 months [45], although effects may last up to 9 months [42,55]. Transient urinary retention following BoNT injection of the detrusor muscle is a potential complication and may require intermittent or indwelling catheterization [41,45,57]. A 2016 phase 3 clinical trial of abobotulinumtoxinA for the treatment of neurogenic detrusor overactivity appears to have been terminated in 2019 due to low patient recruitment [69].

Interestingly, a recent cross-sectional study of over 300 PD patients found that despite no difference in overall urinary symptom prevalence, men with PD are more likely than women to receive a medication, such as BoNT, for urinary symptom treatment [46]. Miller-Patterson and colleagues suggest that men may be more likely than women to be screened for urinary dysfunction, however, the reasons for this disparity are ultimately unclear [46].

### 3.5. Botulinum Neurotoxin for Pain

Pain is unfortunately a commonly reported symptom in patients with PD [20,52,53,70]. One 2008 study conducted in Norway found that pain was reported by more than 80% of PD patients (*n* = 146), with these patients experiencing significantly more pain than the general population [71]. Pain in PD is usually multifactorial and can include musculoskeletal pain, dystonic pain, neuropathic pain, and central pain [52,53]. The mechanisms of pain in PD are complex and poorly understood [50], but musculoskeletal pain and dystonic pain seem to be the most common etiologies [55], with the former typically related to rigidity or motor fluctuations and the latter more commonly associated with end-of-medication or peak-of-medication dosing [52]. BoNT has been shown to alleviate some PD related pain symptoms, particularly painful dystonia [48].

A small randomized controlled crossover study in 2017 showed a mild but non-significant reduction in musculoskeletal and dystonic pain 4 weeks after treatment with BoNT type A, with even greater but still non-significant reduction in dystonic pain on subgroup analysis [47]. In another randomized placebo-controlled trial, injection of incobotulinumtoxinA with 100 units into either the flexor digitorum brevis or flexor digitorum longus was effective in reducing pain associated with plantar flexion of toe dystonia [49]. Finally, as discussed previously, BoNT can help with painful myotonus or spasms in the esophagus, pylorus, anal sphincter, and detrustor muscle [51].

### 3.6. Botulinum Neurotoxin for Other Indications

It is worth noting that BoNT may be useful for the motor aspects of PD as well [24,25,27,56]. BoNT is increasingly being used for additional PD-related indications including blepharospasm (i.e., involuntary eyelid closure), oromandibular dystonia, cervical dystonia, limb dystonia, and tremors [27,56]. Although more common in atypical parkinsonism, blepharospasm can be seen in patients with idiopathic PD [72,73]. BoNT injection of the orbicularis oculi is effective and represents the first-line therapy for blepharospasm [60,74].

## 4. Conclusions

This review has highlighted the various ways in which BoNT can be used to treat the autonomic symptoms of PD. BoNT is effective in the treatment of sialorrhea, hyperhidrosis, gastrointestinal dysfunction, and urinary dysfunction. Other indications for BoNT in PD include pain and motor symptoms such as blepharospasm, oromandibular dystonia, cervical dystonia, limb dystonia, and tremors. There seems to be a relative paucity of primary research on the use of BoNT for autonomic symptoms in the last 4 years. This review illuminates an opportunity for future research on the subject. Future research may also help elucidate the various subtypes of PD and assist with better tailoring treatment. Nevertheless, BoNT is likely to remain a mainstay in the regimen, including in the treatment of autonomic symptoms.

## 5. Methods

The purpose of this systematic review was to explore the varied uses of BoNT for autonomic dysfunction in PD. This review was conducted in accordance with the Preferred Reporting Items for Systematic Reviews and Meta-Analyses (PRISMA) guidelines [29]. For inclusion in this systematic review, studies had to be peer-reviewed, published in the year 2017 or later, and discuss treatment of autonomic symptoms in PD with BoNT.

Systematic literature searches were conducted in three databases, including Cochrane Library, EMBASE, and PubMed. Database searches were conducted between September and November 2020, with article years restricted to 2017 or later. The following combination of search terms were used using the Boolean operator AND: “autonomic AND Botox AND Parkinson’s disease”, “autonomic AND botulinum toxin AND Parkinson’s disease”, “non-motor AND Botox AND Parkinson’s disease”, “non-motor AND botulinum toxin AND Parkinson’s disease”.

Citations and abstracts of the articles found through the databases were hand searched to extract all relevant articles. One author (S.D.M.) independently reviewed the full list of articles to identify which met qualifying criteria. Data extracted from each article included (1) article title, author, and publication year; (2) study methodology; (3) BoNT type (if applicable); (4) number of participants (if applicable); and (5) results. To summarize the findings, systematic methods were utilized.

## Figures and Tables

**Table 1 toxins-13-00226-t001:** Summary of botulinum neurotoxin formulations currently available for clinical use. Adapted from a review by Shukla and Malaty [27].

Toxin Property	OnabotulinumtoxinA	AbobotulinumtoxinA	RimabotulinumtoxinB	IncobotulinmtoxinA
Year introduced	1989	1991	2000	2005
Trade name	BOTOX^®^	Dysport^®^	Myobloc/NeuroBloc^®^	Xeomin^®^
Mechanism of action	Cleaves SNAP 25	Cleaves SNAP 25	Cleaves VAMP	Cleaves SNAP 25
Molecular weight (kD)	900	500–900	700	150
Total protein (ng/vial)	~5	~5	~50	~0.6
Units/vial	50, 100, or 200	300 or 500	2500, 5000, or 10,000	50 or 100
Shelf life (months)	36	24	24	36
Formulation	Vacuum dried	Freeze dried	Sterile solution	Freeze dried
pH after reconstitution	7.4	7.4	5.6	7.4
FDA-approved uses (in adults unless indicated)	Cervical dystonia (16 years and up), blepharospasm (12 years and up), hyperactive bladder, upper and lower limb spasticity, strabismus (12 years and up), glabellar lines, axillary hyperhidrosis, and chronic migraine	Cervical dystonia, glabellar lines, upper limb spasticity, and lower limb spasticity (2 years and up)	Cervical dystonia and chronic sialorrhea	Cervical dystonia, chronic sialorrhea, blepharospasm, and upper limb spasticity (2 years and up)
Off-label uses	Sialorrhea, hemifacialspasm, focal limb dystonia, oromandibular dystonia, tremors, tics, and tardive dyskinesia	Sialorrhea, focal limb dystonia, oromandibular dystonia, and tremors	Focal limb dystonia and oromandibular dystonia	Focal limb dystonia and oromandibular dystonia

Abbreviations: FDA = Food and Drug Administration, SNAP 25 = synaptosomal-associated protein 25, VAMP = vesicle associated membrane protein.

## Data Availability

All data underlying the results are available as part of the article and no additional source data are required.

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
