# Peer review of "Therapeutic Applications of Botulinum Neurotoxin for Autonomic Symptoms in Parkinson’s Disease: An Updated Review"

_toxins, 2021, doi:10.3390/toxins13030226_

Round 1
Reviewer 1 Report
It is a useful review summarizing current evidence for the use of botulinum neurotoxins in treatments of Parkinson's disease. One caveat to address is to list in the Results table the types of BoNT used - type A or type B (or both) - in the Toxin type column. In addition, most of the next column listing number of patients consists of N/A which doesn't look presentable. The authors could add the few available numbers in another column.
Reviewer 2 Report
The article "Therapeutic applications of botulinum toxin type A in Parkinson's disease" is a well written informative short review of the recent literature. It also warns the readers on the possible novel indications in the e.g. treatment of movement disorders such as blepahrospasm which may be present in PD. The literature list is somewhat limited to the more recent reviews, rather than citing original clinical studies. Other than that I have no further comments.
